 

# Six-year retrospective analysis of the epidemiology and risk factors of multidrug-resistant bloodstream infections in oncology patients in Jiangxi, China

Bin Xu,[1] Xunsong Wang,[2] Xiaohui Li,[1] Muhammad Nadeem Khan,[3] Muhammad Shafiq,[4] Sabir Khan,[5] Tenghua Yu,[1] Rahat Ullah Khan,[6,7] Ying Song,[2] Hanman Qiu,[1] Qiao-Li Lv,[1] Hazrat Bilal[1]

**ABSTRACT**   Cancer patients are particularly vulnerable to bloodstream infections (BSIs) due to their immunocompromised status. This retrospective study at Jiangxi Cancer Hospital (2019 to 2024) analyzed the bacterial spectrum, risk factors, and antimicrobial susceptibility profiles of BSIs in cancer patients. Logistic regression analysis was performed to identify risk factors associated with multidrug resistance (MDR) in Gram-positive and Gram-negative infections. A total of 1,161 bacteria were isolated from 954 oncological patients in 1,095 BSI episodes. Among these, 62.79% were Gram-negative bacteria, with *Escherichia coli* (40.88%) and *Klebsiella pneumoniae* (19.07%) being the most prevalent. Gram-positive bacteria accounted for 37.21% of isolates, with *Staphylococcus aureus* (26.16%) being the most common. Gram-negative infections were more prevalent in patients with breast, gastrointestinal, hepatobiliary, and pancreatic cancers, whereas Gram-positive infections were more common among those with lymphoma, head and neck, and lung cancers, as well as in patients undergoing invasive procedures. High resistance rates were observed against ampicillin, piperacillin, cefazolin, and erythromycin, whereas tigecycline and nitrofurantoin exhibited low resistance rates among the tested bacterial isolates. Among Gram-negative bacteria, 5.48% were carbapenemase producers and 15.36% were extended-spectrum β-lactamase (ESBL) producers. Moreover, methicillin resistance was observed in 36.04% of *Staphylococcus aureus* and 66.83% of coagulase-negative *Staphylococcus* isolates. MDR was observed in 40.19% of Gram-negative and 50.92% of Gram-positive infections. Hypoproteinemia (OR 3.2) was identified as an independent risk factor for MDR-BSIs caused by Gram-negative bacteria. The high prevalence of MDR bacteria in BSIs among cancer patients highlights the necessity of individualized treatment and continuous monitoring in oncology settings.

**IMPORTANCE**   This study addresses a critical gap in understanding the epidemiology and risk factors of multidrug-resistant bloodstream infections (MDR-BSIs) in oncology patients from a high-burden region of China. By analyzing over a thousand BSI episodes over 6 years, the study identifies the predominant bacterial species, their resistance profiles, and key clinical and procedural risk factors associated with MDR. The high rates of MDR among both Gram-negative and Gram-positive pathogens underscore the urgent need for tailored antimicrobial stewardship, infection control interventions, and individualized treatment strategies. These findings provide essential evidence for guiding empirical antibiotic choices and improving infection management protocols in cancer care settings.

**KEYWORDS**   bloodstream infections, cancer, multidrug resistance, risk factor, inappropriate empirical therapy

**Peer Reviewer** Gabriele Giuliano, University of Siena, Siena, Italy

Address correspondence to Hazrat Bilal, bilal.microbiologist@yahoo.com, or Qiao-Li Lv, lvqiaoli2008@126.com.

The authors declare no conflict of interest.

Cancer patients are often immunocompromised, rendering them susceptible to bloodstream infections (BSIs), which significantly increase morbidity and mortality. The risk of BSI in cancer patients is increased by invasive treatments and procedures such as chemotherapy, central venous catheterization (CVC), and surgeries (1). Both Gram-positive and Gram-negative bacteria can cause BSIs and are generally treatable with antibiotics. However, the widespread overuse of antibiotics has led to the emergence of resistant bacterial strains. In oncological settings, multidrug-resistant (MDR) bacterial BSIs present a serious challenge, leading to higher treatment failure rates, prolonged hospital stays, and increased healthcare costs (2).

BSIs in China are associated with high morbidity and mortality among cancer patients. Recent studies have reported that the incidence of BSIs in hospitalized cancer patients ranges from 6.5% to 30%, with mortality rates reaching 18% to 28% in some cases (3–5). This situation is worsened by the rise of MDR pathogens, with over 50% of *Escherichia coli* and 20% of *Klebsiella pneumoniae* isolates producing extended-spectrum beta-lactamase (ESBL) (6). Moreover, high rates of methicillin resistance have been reported in coagulase-negative *staphylococci* (MRCoNS) and *Staphylococcus aureus* (MRSA), at 84.7% and 48.9%, respectively. These figures underscore the growing threat of antimicrobial resistance in this vulnerable population, further complicating clinical management (6, 7).

The epidemiology of BSIs varies geographically, influenced by patients' demographics, clinical practices, and local antimicrobial susceptibility trends (8). However, data on BSIs in cancer patients, a high-risk group, remain limited (9). Given the rising incidence of cancer cases in China, a comprehensive analysis of bacterial BSIs and their resistant profiles is necessary (10). This retrospective study conducted at Jiangxi Cancer Hospital analyzed risk factors, bacterial spectrum, and antimicrobial susceptibility in cancer patients with BSIs over 6 years. The findings aim to optimize infection control practices, guide empirical antibiotic therapy, and improve patient outcomes.

## MATERIALS AND METHODS

### Study setting and data collection

This study was conducted at Jiangxi Cancer Hospital, a tertiary A-level institution located in Nanchang City, Jiangxi Province, China. With 2,030 beds, the hospital serves a population of 45 million, providing care to urban and rural patients throughout the province. Demographic and clinical data related to BSI were retrospectively collected from electronic records over 6 years (from January 2019 to December 2024). All inpatients diagnosed with BSI were included. Collected data encompassed patient age, gender, socioeconomic status, cancer type, cancer therapy, procedures and interventions during hospitalization, complications, length of hospital stay (days), and survival status within 30 days of BSI diagnosis. Additionally, antibiotic susceptibility testing (AST) data for all isolates were obtained from the hospital's microbiology laboratory.

### Definitions

BSI was defined as a positive blood culture accompanied by a high fever (>38°C) and one or more symptoms such as chills, tachycardia, tachypnea, hypotension, fatigue, nausea or vomiting, and altered mental status (11). For commensal bacteria like CoNS, at least two consecutive positive blood cultures are required to confirm BSI. BSIs were classified as polymicrobial if two or more bacterial isolates were identified and monomicrobial if only one species was detected. Hospital-acquired BSIs are those diagnosed more than 72 h after admission, whereas community-acquired BSIs are identified within 72 h of admission (12). Complications of BSI included sepsis, septic shock, organ dysfunctions such as respiratory failure and electrolyte imbalances, vascular complications like thrombosis, and treatment-related conditions, including myelosuppression and neutropenia. Co-infections or potential sources of BSI included pneumonia, urinary tract

infections (UTI), and biliary tract infections. An absolute neutrophil count below 0.5 × $10^9$/L is defined as neutropenia, while a count exceeding 7.5 × $10^9$/L is considered neutrophilia. Bacteria are classified as MDR if they are resistant to at least one antibiotic in three or more different classes (13).

## Microbiological methods

Blood samples were routinely collected based on clinical indications and processed using the Versa TREK-6240 Automated Blood Culture System (UK). Positive samples were subcultured on selective media, including Columbia agar, Chocolate agar, and MacConkey agar plates, and incubated for 24 to 48 h. Bacterial identification was performed using matrix-assisted laser desorption/ionization time-of-flight mass spectrometry Smart 5020. Matrix solution and sample processing reagents for microbial mass spectrometry were purchased from Zhuhai Deere Bioengineering Co., Ltd. (Zhuhai, Guangdong, China). AST was conducted using the Sensititre ARIS-2X system (Thermo Fisher). Sensitivity results were primarily interpreted according to CLSI guidelines for all tested antibiotics (14), except for tigecycline, which was interpreted according to EUCAST (2024) breakpoints (http://www.eucast.org/clinical_breakpoints/). ESBL, carbapenemase production, and methicillin resistance were suspected from MIC results and confirmed using the double disk synergy test, modified Hodge test, and cefoxitin disc diffusion test, respectively (15, 16).

## Data analysis

Data collected in Excel were categorized into Gram-positive and Gram-negative bacteria, with each group further classified as MDR and non-MDR based on AST profiles. Categorical variables were presented as total numbers and percentages, while continuous variables were reported as median and interquartile range (IQR). The chi-square test was used for categorical variables, and the Mann-Whitney U test was applied for continuous variables. Risk factor analysis was conducted using univariate and multivariate logistic regression to identify variables associated with MDR-BSIs in cancer patients. Variables with a $P$ value <0.10 in the univariate analysis were included in the multivariate logistic regression model. A $P$ value <0.05 was considered statistically significant. Graphical representations and descriptive statistics were conducted using GraphPad Prism ($v$. 9), while the logistic regression analyses were performed using R ($v$. 4.4.1).

## Study aims

The primary aim of this study was to identify risk factors associated with MDR-BSIs among cancer patients. The secondary aims included (i) comparison of demographic, clinical, and treatment characteristics between Gram-negative and Gram-positive bacteria, (ii) analysis of antibiotic susceptibility profiles, and (iii) 30-day mortality following BSI diagnosis.

## RESULTS

### Overview of BSI episodes and bacterial isolates

During the study period, 1,095 BSI episodes were recorded among 954 oncological patients. Of these episodes, 1,032 were monomicrobial, while 63 were polymicrobial. Among the polymicrobial cases, each of the 60 involved two bacterial species, and three cases contained three bacterial species each. The distribution of episodes among patients is presented in Table S1 (supplementary file). Bacterial species identified in polymicrobial BSI episodes are listed in Table S2 (supplementary file).

A total of 1,161 bacteria were reported: 729 (62.79%) Gram-negative and 432 (37.21%) Gram-positive. Among the Gram-negative bacteria, *E. coli* was most prevalent ($n = 298$, 40.88%), followed by *K. pneumoniae* ($n = 139$, 19.07%) and *Enterobacter cloacae* ($n = 72$, 9.88%). Among Gram-positive bacteria, *S. aureus* was most common ($n = 113$, 26.16%), followed by *Staphylococcus epidermidis* ($n = 68$, 15.74%) and *Staphylococcus hominis* ($n =$

62, 14.35%). The counts and percentages of all reported bacteria are detailed in Fig. S1 (supplementary file).

Regarding cancer types, Gram-positive bacteria were significantly more common than Gram-negative bacteria in lymphoma (14.35% vs 8.50%, $P = 0.001$), lung (13.65% vs 9.87%, $P = 0.04$), and head and neck cancers (11.80% vs 5.76%, $P < 0.001$). In contrast, Gram-negative bacteria were significantly more prevalent than Gram-positive bacteria in breast (10.56% vs 6.71%, $P = 0.02$), hepatobiliary (20.16% vs 13.19%, $P = 0.002$), pancreatic (5.07% vs 2.55%, $P = 0.03$), and gastrointestinal cancers (22.91% vs 15.97%, $P = 0.004$) (Fig. 1).

## Comparative analysis of demographic and clinical characteristics between Gram-negative and Gram-positive bacterial cases

The clinical and demographic characteristics, along with a comparative analysis between Gram-negative and Gram-positive bacterial cases, are presented in Table S3 (supplementary file). The distribution of BSIs was slightly higher in males ($n = 676$, 58.23%) than in females ($n = 485$, 41.77%). Regarding bacterial types, Gram-negative BSIs were slightly more prevalent in both male ($n = 410$, 60.65%) and female ($n = 319$, 65.77%) populations compared to Gram-positive infections, although the difference was not statistically significant. The median age of all cases was 59 years (IQR: 50–67). Patients with Gram-negative infections had a slightly higher median age of 59 years (IQR: 50–67) compared

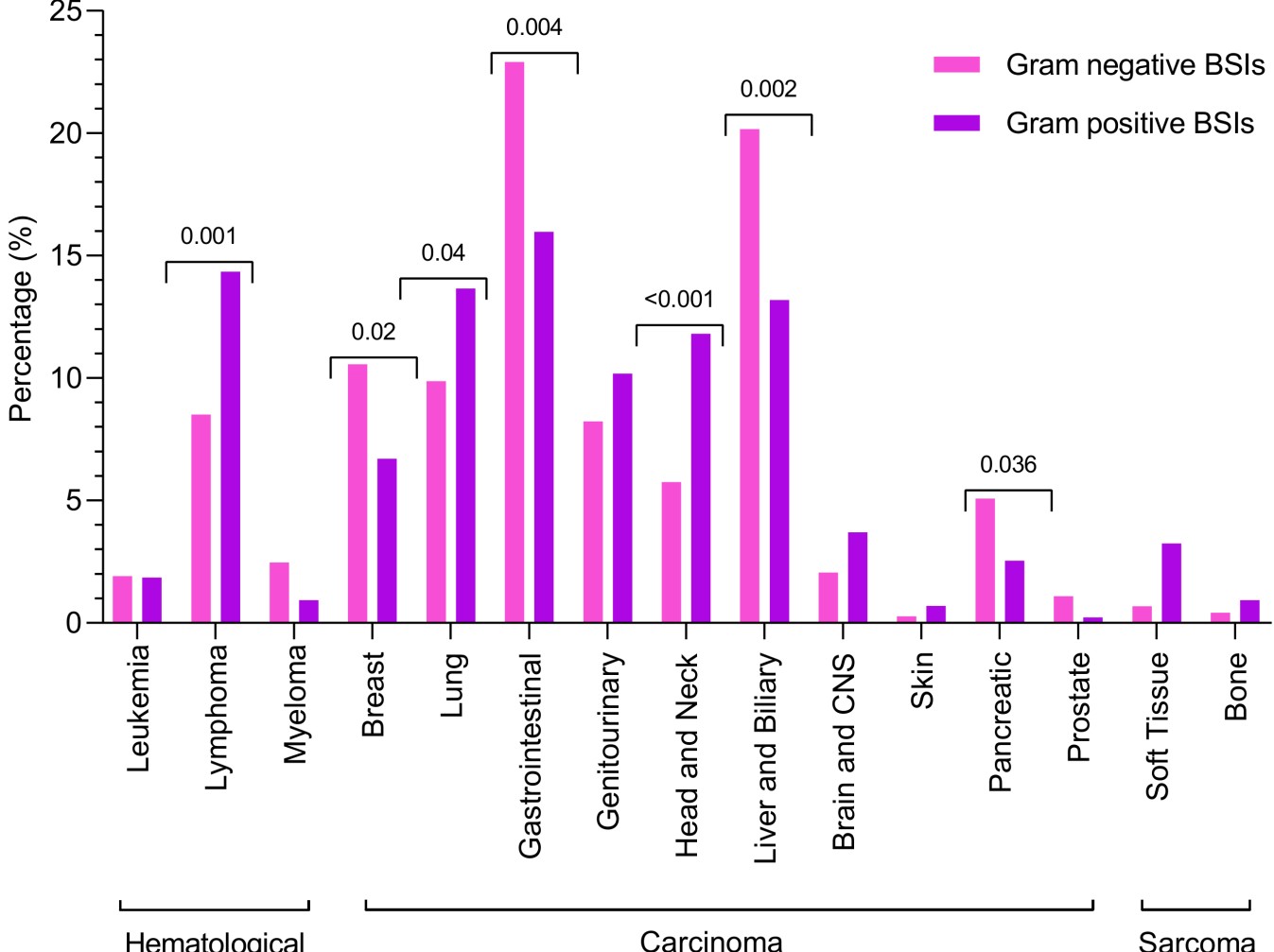

FIG 1 Comparative analysis of Gram-positive and Gram-negative bacteria in various cancer types.

to those with Gram-positive infections (58 years, IQR: 49–67). Most cases were reported among laborers or manual workers ($n = 650$, 55.99%), followed by unemployed or retired individuals ($n = 349$, 30.06%). Most BSI cases were reported in individuals with a basic level of education ($n = 386$, 33.25%), followed by those with no formal education ($n = 317$, 27.3%), while the population with higher education had the lowest proportion ($n = 68$, 5.86%). Among individuals with a basic level of education, Gram-negative infections ($n = 258$, 66.84%) were significantly more common than Gram-positive infections ($n = 128$, 33.16%) ($P = 0.04$).

Of the infections, 950 (81.83%) cases were hospital-acquired, while 211 (18.17%) were community-acquired. Both hospital-acquired and community-acquired cases were significantly more common in Gram-negative BSIs than in Gram-positive BSIs ($P < 0.001$). Among various cancer treatments, the highest proportion of cases was observed in patients undergoing chemotherapy ($n = 669$, 57.62%), while the lowest was in those receiving immunotherapy ($n = 102$, 8.79%). The difference between Gram-positive and Gram-negative bacterial groups across various cancer treatments was not statistically significant. Medical procedures and interventions, such as ventilators, intra-abdominal catheters, nasogastric feeding, and surgical interventions, were significantly more common in the Gram-positive group compared to the Gram-negative group ($P < 0.05$ for all). Regarding infection-related variables, sepsis occurred in 570 cases (49.1%), while septic shock was reported in 82 (7.06%) cases. Notably, Gram-negative bacteria were significantly more prevalent than Gram-positive infections in cases of UTIs, septic shock, and biliary tract infections ($P \leq 0.01$).

Among other conditions and complications, neutrophilia ($n = 475$, 40.91%) was the most common, followed by hypoproteinemia ($n = 429$, 36.95%), anemia ($n = 325$, 27.99%), myelosuppression ($n = 306$, 26.36%), and electrolyte disorders ($n = 222$, 19.12%). Notably, hypoproteinemia and neutrophilia were significantly more common in Gram-negative infections, while thrombosis was more prevalent in Gram-positive infections ($P < 0.05$). The median hospital stay for all BSI cases was 20 days (IQR: 13–31). The 30-day mortality rate was 5% ($n = 58$), with 35 (60.34%) from the Gram-negative group and 23 (39.66%) from the Gram-positive group.

## Antibiotic susceptibility profiles

### AST of Gram-negative bacteria

A total of 25 antibiotics were tested against Gram-negative bacteria. The susceptibility profiles of *E. coli*, *Klebsiella* species, and *Enterobacter* species are presented in Table S4 (supplementary file), while those of *Acinetobacter* species, *Pseudomonas* species, and other Gram-negative bacteria are detailed in Table S5 (supplementary file). The percentages and counts of non-susceptible Gram-negative isolates are shown in Fig. 2. Among *E. coli* isolates, high resistance rates were observed to ampicillin ($n = 180$, 88.24%), followed by piperacillin ($n = 136$, 67.66%), tetracycline ($n = 186$, 62.84%), cefazolin ($n = 172$, 58.11%), ciprofloxacin ($n = 169$, 56.9%), cefuroxime ($n = 124$, 55.86%), ceftriaxone ($n = 161$, 54.39%), trimethoprim/sulfamethoxazole ($n = 161$, 54.39%), and levofloxacin ($n = 158$, 53.38%). None of the tested *E. coli* were resistant to tigecycline, and only one (0.34%) isolate was resistant to nitrofurantoin. Among the tested *E. coli* isolates, 62 (20.95%) were ESBL positive, 8 (2.69%) were carbapenemase producers, and 187 (62.75%) were classified as MDR.

Among *Klebsiella* species, a high resistance rate to ampicillin ($n = 117$, 88.64%) was observed, attributable to intrinsic resistance. Considerable resistance was also noted against cefazolin ($n = 75$, 42.37%), cefuroxime ($n = 44$, 32.59%), and ampicillin/sulbactam ($n = 41$, 31.54%). Conversely, resistance was lowest against nitrofurantoin ($n = 3$, 1.69%) and tigecycline ($n = 1$, 0.56%). Among the 181 *Klebsiella* isolates, 33 (18.64%) were ESBL positive, 14 (7.87%) were carbapenemase producers, and 54 (29.83%) were classified as MDR.

Among *Enterobacter* species, high resistance rates were observed to cefazolin ($n = 73$, 91.25%) and ampicillin ($n = 52$, 91.23%), while 100% were susceptible to tigecycline.

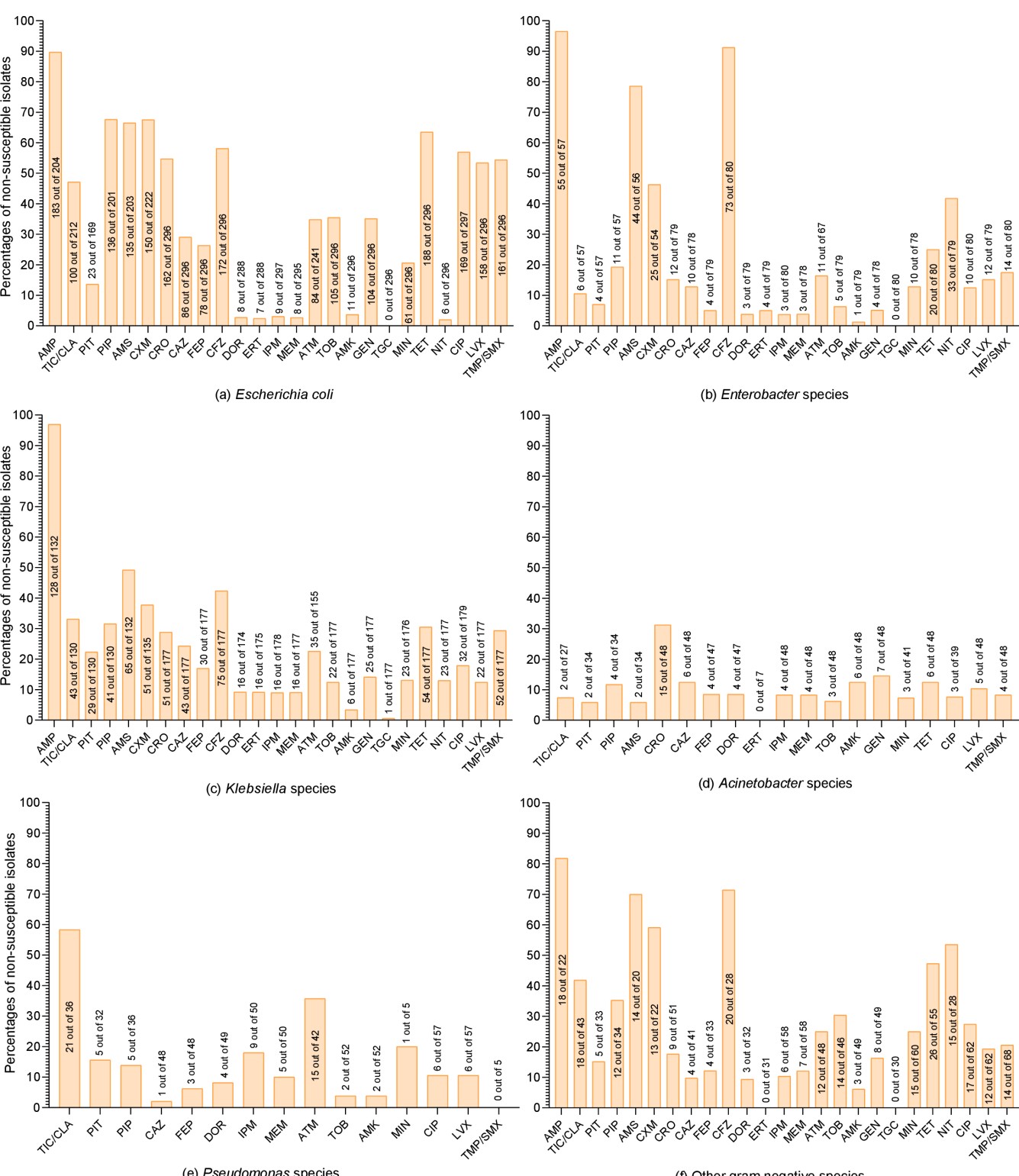

**FIG 2** Non-susceptible Gram-negative isolates in the current study. (a) *Escherichia coli*; (b) *Enterobacter* species; (c) *Klebsiella* species; (d) *Acinetobacter* species; (e) *Pseudomonas* species; (f) other gram negative species. On the y-axis of each bar, the percentages are presented, while in each bar, the number of non-susceptible isolates out of the tested isolates is written. On the x-axis, the names of tested antibiotics are presented, which are Amikacin (AMK), Ampicillin (AMP), Ampicillin/Sulbactam (AMP/SUL), Aztreonam (ATM), Cefazolin (CFZ), Cefepime (CFP), Ceftazidime (CAZ), Ceftriaxone (CRO), Cefuroxime (CXM), Ciprofloxacin (CIP), Doripenem (DOR), Ertapenem (ERT), Gentamicin (GEN), Imipenem (IPM), Levofloxacin (LEV), Meropenem (MEM), Minocycline (MIN), Nitrofurantoin (NIT), Piperacillin (PIP), Piperacillin/Tazobactam (PIP/TAZ), Tetracycline (TET), Ticarcillin/Clavulanic Acid (TIC/CLA), Tigecycline (TGC), Tobramycin (TOB), and Trimethoprim/Sulfamethoxazole (SXT).

Of the 78 tested *Enterobacter* isolates, 7 (8.97%) were ESBL positive, 2 (2.5%) were carbapenemase producers, and 21 (26.92%) were MDR. Among the 48 tested *Acineto-bacter* isolates, the highest resistance was reported against ceftriaxone ($n = 6$, 12.5%). Additionally, five isolates (10.42%) were ESBL-positive, four (8.33%) were carbapenemase producers, and five (10.42%) were classified as MDR. Among *Pseudomonas* species, the highest resistance was observed against ticarcillin/clavulanic acid ($n = 8$ out of 36, 22.22%), followed by aztreonam ($n = 8$, 19.05%). Only one isolate (2.08%) was an ESBL-positive, six (12%) were carbapenemase producers, and one out of 52 (1.92%) tested *Pseudomonas* species was MDR. Among other Gram-negative species, four isolates were ESBL positive, six carbapenemase producers, and 24 out of 72 (33.34%) were classified as MDR.

### AST of Gram-positive bacteria

Nineteen antibiotics were tested against Gram-positive bacteria. The susceptibility profile of *S. aureus,* CoNS, *Enterococcus* species*,* and *Streptococcus* species is detailed in Tables S6 and S7 (supplementary file), with the non-susceptibility percentages depicted in Fig. 3. Among the tested *S. aureus* isolates, high resistance rates were noted against ampicillin and penicillin G ($n = 101$, 90.18% each), followed by oxacillin ($n = 40$, 36.04%), erythromycin ($n = 50$, 44.64%), ciprofloxacin ($n = 23$, 20.72%), and levofloxacin ($n = 18$, 16.07%). None of the tested *S. aureus* isolates were resistant to nitrofurantoin, vancomycin, daptomycin, and linezolid. Thirty-five (30.97%) isolates were MDR, and 40 (36.04%) were MRSA. Among the tested CoNS isolates, a high resistance rate was observed against penicillin G ($n = 179$, 88.18%), followed by ampicillin ($n = 177$, 87.19%), and erythromycin ($n = 144$, 70.59%). Tigecycline, daptomycin, linezolid, and quinupristin/dalfopristin were among the most effective antibiotics, showing 100% susceptibility against the tested CoNS. Of the CoNS isolates, 126 (60.58%) were MDR, while 135 (66.83%) were MRCoNS.

Among the 58 tested *Enterococcus* isolates, 48 (82.75%) were MDR. The highest resistance was reported against levofloxacin ($n = 36$, 62.07%) and ciprofloxacin ($n = 37$, 66.07%), while vancomycin was the most effective drug, with no resistance observed. *Streptococcus* species were 100% susceptible to tigecycline, vancomycin, chloramphenicol, daptomycin, and linezolid. Among 43 tested *Streptococcus* isolates, nine (20.93%) were MDR, with the highest resistance observed against erythromycin ($n = 18$, 42.86%) and tetracycline ($n = 12$, 35.29%). Among the other Gram-positive isolates, one *Corynebacterium jeikeium* and one *Corynebacterium striatum* isolate were MDR.

### MDR versus non-MDR

An analysis of MDR versus non-MDR cases across various cancer types was performed. Among Gram-negative BSIs, MDR cases were significantly more common in brain and prostate cancers, whereas non-MDR cases were more prevalent in lymphoma and lung cancers ($P < 0.05$). In contrast, among Gram-positive BSIs, the distribution of MDR and non-MDR cases across cancer types was not statistically significant (Fig. 4). Risk factor analysis of MDR versus non-MDR cases was performed, as shown in Table S8 (supplementary file). To enhance clarity, separate analyses for Gram-negative and Gram-positive BSIs were also conducted and are presented in the following sections.

### Risk factors for MDR in Gram-negative bacterial BSIs

The risk factor analysis for Gram-negative bacterial infections is presented in Table 1. The distribution of MDR and non-MDR cases between males and females was not statistically significant. The median age of the MDR group was significantly higher at 61 years (IQR: 52 to 69) compared to the non-MDR group, which had a median age of 58 years (IQR: 49 to 66) ($P < 0.001$). In the various occupation groups, the MDR rate was slightly higher in the laborer and manual worker group, with an odds ratio (OR) of 1.21 (95% CI: 0.89 to 1.63); however, the difference was not statistically significant ($P = 0.21$). Across the different education levels, infection sources, and cancer treatment groups, no significant differences were observed between MDR and non-MDR Gram-negative BSIs.

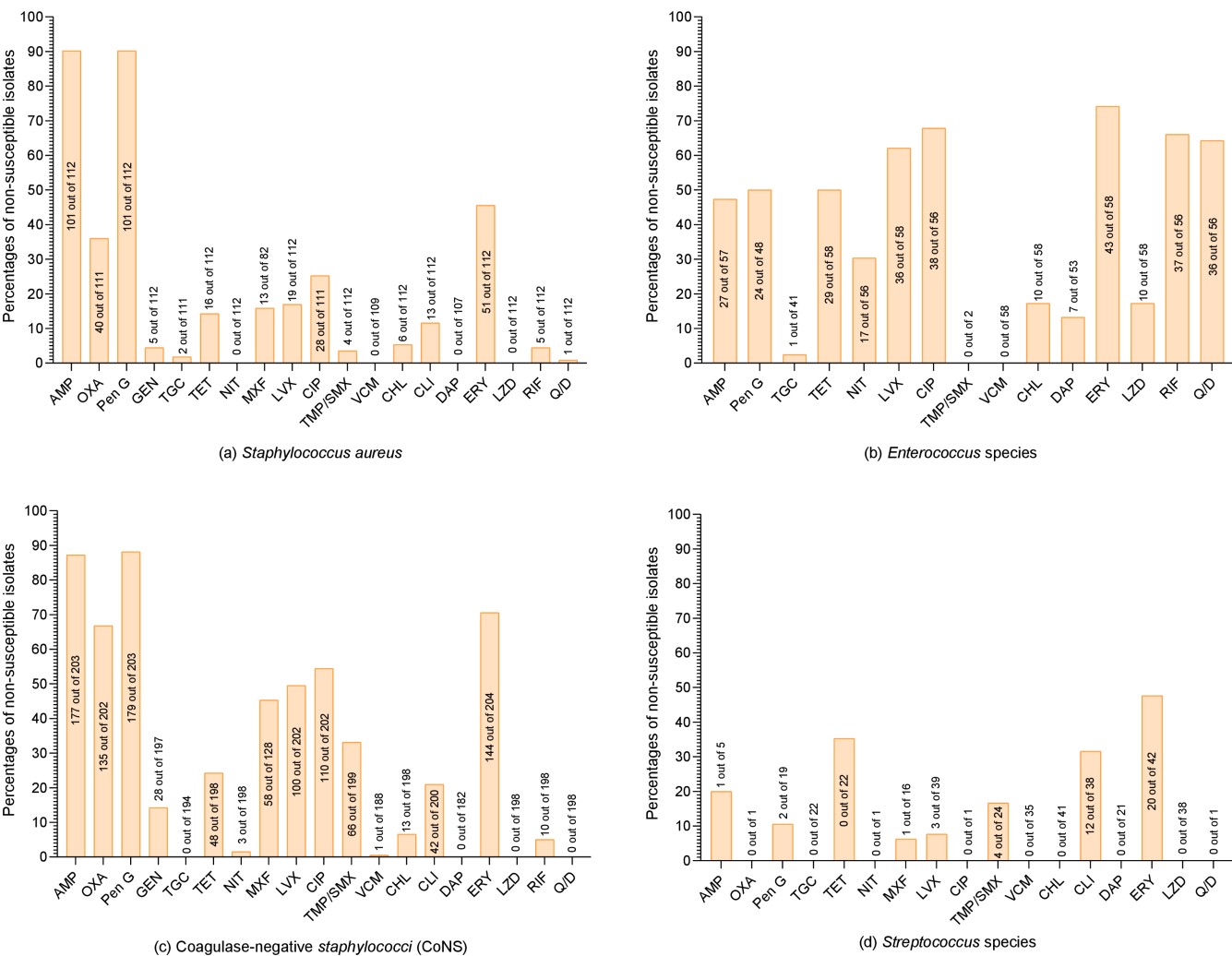

**FIG 3** Non-susceptible Gram-positive isolate in the current study. (a) *Staphylococcus aureus*; (b) *Enterococcus* species; (c) coagulase-negative *Staphylococci* (CoNS); (d) *Streptococcus* species. On the y-axis of each bar, the percentages are presented, while in each bar, the number of non-susceptible isolates out of the tested isolates is written. On the x-axis, the names of tested antibiotics are presented, which are Ampicillin (AMP), Chloramphenicol (CHL), Ciprofloxacin (CIP), Clindamycin (CLI), Daptomycin (DAP), Erythromycin (ERY), Gentamicin (GEN), Levofloxacin (LEV), Linezolid (LZD), Moxifloxacin (MXF), Nitrofurantoin (NIT), Oxacillin (OXA), Penicillin G (PEN), Quinupristin/Dalfopristin (Q/D), Rifampin (RIF), Tetracycline (TET), Tigecycline (TGC), Trimethoprim/Sulfamethoxazole (SXT), and Vancomycin (VAN).

Regarding hospital procedures and interventions, patients who used urinary catheters and received parenteral nutrition were at a higher risk of MDR, with an OR (95% CI) of 1.91 (1.34 to 2.71) ($P < 0.001$) and 1.56 (1.13 to 2.15) ($P = 0.01$), respectively. Concerning co-infectious diseases, UTIs were more prevalent in the MDR group (11.95%) compared to the non-MDR group (5.05%), with an odds ratio (95% CI) of 2.55 (1.48 to 4.50) ($P < 0.001$). Notably, hypoproteinemia was significantly more prevalent in patients with MDR infections (OR [95% CI]: 1.59 [1.17 to 2.14], $P = 0.002$). No statistically significant differences were observed between MDR and non-MDR BSI cases regarding hospital stay ($P = 0.06$). However, the 30-day mortality rate for the MDR group was substantially greater than for the non-MDR group (6.83% vs 3.44%), with an OR (95% CI) of 2.06 (1.05 to 3.99) ($P = 0.04$).

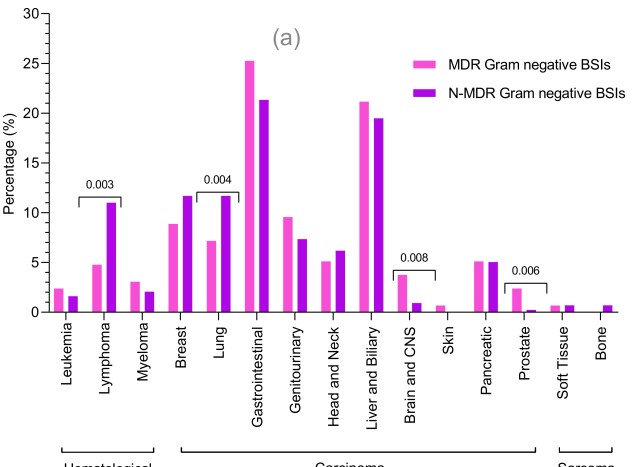
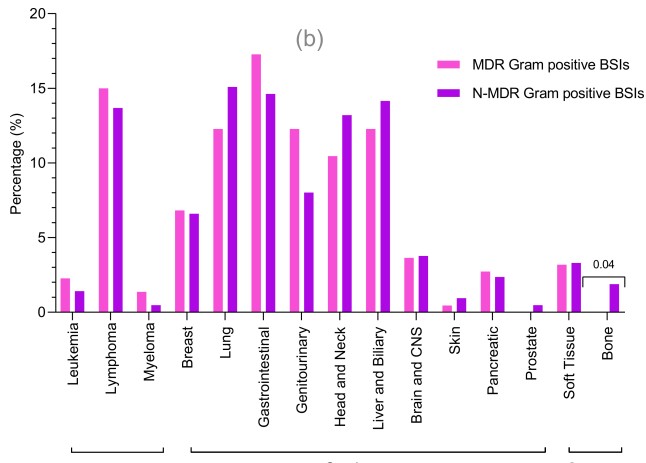

**FIG 4** Distribution of MDR versus N-MDR BSIs in cancer patients. (a) Comparison of MDR and N-MDR Gram-negative bacteria in various cancer types. (b) Comparison of MDR and N-MDR Gram-positive bacteria in various cancer types.

## Risk factors for MDR in Gram-positive bacterial BSIs

The risk factor analysis of Gram-positive bacteria for MDR versus non-MDR is presented in Table 2. The demographics, socioeconomic level, occupations, and source of infection of the patients did not significantly differ between the MDR and non-MDR groups. Regarding cancer treatments, the proportion of non-MDR cases was slightly higher than that of MDR cases; however, the differences were not statistically significant for chemotherapy, radiotherapy, and targeted therapy ($P > 0.05$). In contrast, among immunotherapy-treated patients, non-MDR cases were significantly more prevalent than MDR cases (10.85% vs 4.55%, $P = 0.01$).

The use of a urinary catheter and nasogastric feeding was significantly linked to the prevalence of MDR than non-MDR, with OR (95% CI) of 1.65 (1.06 to 2.57) ($P = 0.03$) and 2.05 (1.02 to 4.25) ($P = 0.04$), respectively. The proportion of UTIs was significantly more prevalent in the MDR group (5.91%) than in the non-MDR group (1.42%), with an OR (95% CI) of 4.38 (1.26 to 14.57) ($P = 0.01$). Regarding comorbidities and complications, nausea and vomiting were significantly associated with the MDR group compared to the non-MDR group ($P = 0.04$). However, the overall proportion of nausea and vomiting in Gram-positive cases was low (11 out of 432, 2.54%). Moreover, patients with thrombosis had a lower proportion of MDR compared to non-MDR (2.73% vs 8.96%) ($P = 0.01$). For all other conditions and complications, the differences between MDR and non-MDR were statistically insignificant. Similarly, no statistically significant differences were observed between the MDR and non-MDR groups regarding hospital stay and 30-day mortality rates ($P > 0.05$).

## Multivariate regression analysis

Furthermore, multivariate regression analysis was conducted after identifying significant variables in the univariate analysis. This was intended to clarify independent associations between potential risk factors and MDR-BSIs in cancer patients. For MDR Gram-positive BSIs, the analysis yielded extremely wide confidence intervals due to the limited number of events, rendering the results uninterpretable. In contrast, for Gram-negative BSI, hypoproteinemia was significantly associated with an increased risk of MDR (OR [95% CI]): 16.23 (6.66–46, $P < 0.001$) (Table 3).

## DISCUSSION

This study presents a comprehensive analysis of risk factors and antimicrobial-resistant patterns of bacterial BSIs among cancer patients in Jiangxi, China. A total of

**TABLE 1** Univariate analysis of demographic, clinical, and treatment variables between MDR and non-MDR Gram-negative bacterial BSI in cancer patients

| Variables | MDR (n = 293) | MDR (%) (IQR) | N-MDR (n = 436) | N-MDR (%) (IQR) | Odds ratio | 95% CI | P value |
|---|---|---|---|---|---|---|---|
| Gender | | | | | | | |
| Male | 174 | 59.39 | 236 | 54.13 | 1.24 | 0.92 to 1.66 | 0.16 |
| Female | 119 | 40.61 | 200 | 45.87 | 0.81 | 0.60 to 1.09 | 0.16 |
| Age (median [IQR]) | 61 | (52 to 69) | 58 | (49 to 66) | −3[b] | | **<0.001[a]** |
| Occupation | | | | | | | |
| Jobless/unemployed/retired | 82 | 27.99 | 130 | 29.82 | 0.91 | 0.66 to 1.27 | 0.59 |
| Laborer/manual worker | 177 | 60.41 | 243 | 55.73 | 1.21 | 0.89 to 1.63 | 0.21 |
| Freelancer/self-employed | 15 | 5.12 | 44 | 10.09 | 0.48 | 0.26 to 0.87 | 0.02 |
| Professional and technical | 19 | 6.48 | 19 | 4.36 | 1.52 | 0.81 to 2.86 | 0.21 |
| Education background | | | | | | | |
| No formal education | 80 | 27.3 | 113 | 25.92 | 1.07 | 0.76 to 1.50 | 0.68 |
| Basic education | 104 | 35.49 | 153 | 35.09 | 1.02 | 0.74 to 1.39 | 0.91 |
| Intermediate education | 43 | 14.68 | 85 | 19.5 | 0.71 | 0.48 to 1.06 | 0.09 |
| Secondary education | 46 | 15.7 | 64 | 14.68 | 1.08 | 0.72 to 1.64 | 0.71 |
| Higher education | 20 | 6.83 | 21 | 4.82 | 1.45 | 0.76 to 2.74 | 0.25 |
| Source of infection | | | | | | | |
| Hospital acquired | 235 | 80.2 | 338 | 77.52 | 1.18 | 0.82 to 1.68 | 0.39 |
| Community acquired | 58 | 19.8 | 98 | 22.48 | 0.85 | 0.59 to 1.22 | 0.39 |
| Cancer treatment | | | | | | | |
| Chemotherapy | 162 | 55.29 | 254 | 58.26 | 0.89 | 0.66 to 1.20 | 0.43 |
| Radiotherapy | 34 | 11.6 | 45 | 10.32 | 1.14 | 0.70 to 1.80 | 0.58 |
| Targeted therapy | 54 | 18.43 | 88 | 20.18 | 0.89 | 0.61 to 1.31 | 0.56 |
| Immunotherapy | 35 | 11.95 | 34 | 7.8 | 1.6 | 0.98 to 2.62 | 0.06 |
| Procedures and Interventions | | | | | | | |
| CVC | 38 | 12.97 | 50 | 11.47 | 1.15 | 0.74 to 1.79 | 0.54 |
| Urinary catheter | 85 | 29.01 | 77 | 17.66 | 1.91 | 1.34 to 2.71 | **<0.001[c]** |
| PICC | 51 | 17.41 | 70 | 16.06 | 1.1 | 0.74 to 1.65 | 0.63 |
| Safety IV catheter | 155 | 52.9 | 255 | 58.49 | 0.8 | 0.59 to 1.08 | 0.14 |
| Ventilator | 13 | 4.44 | 9 | 2.06 | 2.2 | 0.93 to 5.47 | 0.07 |
| Blood transfusion | 77 | 26.28 | 101 | 23.17 | 1.18 | 0.84 to 1.67 | 0.34 |
| Biopsies | 33 | 11.26 | 53 | 12.16 | 0.92 | 0.57 to 1.45 | 0.71 |
| Intra-abdominal catheter | 60 | 20.48 | 73 | 16.74 | 1.28 | 0.87 to 1.86 | 0.2 |
| Parenteral nutrition | 105 | 35.84 | 115 | 26.38 | 1.56 | 1.13 to 2.15 | **0.01** |
| Nasogastric feeding | 17 | 5.8 | 18 | 4.13 | 1.43 | 0.71 to 2.83 | 0.3 |
| Surgery | 27 | 9.22 | 30 | 6.88 | 1.37 | 0.79 to 2.36 | 0.25 |
| Infection and infectious diseases | | | | | | | |
| Sepsis | 144 | 49.15 | 217 | 49.77 | 0.98 | 0.72 to 1.31 | 0.87 |
| Septic shock | 30 | 10.24 | 36 | 8.26 | 1.27 | 0.77 to 2.10 | 0.36 |
| Hepatitis B | 43 | 14.68 | 55 | 12.61 | 1.19 | 0.78 to 1.83 | 0.42 |
| Pneumonia | 43 | 14.68 | 66 | 15.14 | 0.96 | 0.63 to 1.47 | 0.86 |
| Urinary tract infection | 35 | 11.95 | 22 | 5.05 | 2.55 | 1.48 to 4.50 | **<0.001** |
| Biliary tract infection | 32 | 10.92 | 42 | 9.63 | 1.15 | 0.71 to 1.87 | 0.57 |
| Conditions and complications | | | | | | | |
| Anemia | 96 | 32.76 | 114 | 26.15 | 1.38 | 0.99 to 1.91 | 0.05 |
| Hypertension | 41 | 13.99 | 69 | 15.83 | 0.87 | 0.58 to 1.32 | 0.5 |
| Diabetes | 30 | 10.24 | 33 | 7.57 | 1.39 | 0.84 to 2.37 | 0.21 |
| Thrombosis | 9 | 3.07 | 6 | 1.38 | 2.27 | 0.85 to 6.37 | 0.11 |
| Myelosuppression | 76 | 25.94 | 109 | 25 | 1.05 | 0.75 to 1.48 | 0.78 |
| Hypoproteinemia | 137 | 46.76 | 155 | 35.55 | 1.59 | 1.17 to 2.14 | **0.002** |
| Nausea and Vomiting | 11 | 3.75 | 16 | 3.67 | 1.02 | 0.45 to 2.24 | 0.95 |
| Radiation dermatitis | 7 | 2.39 | 7 | 1.61 | 1.5 | 0.56 to 4.01 | 0.45 |
| Respiratory failure | 11 | 3.75 | 16 | 3.67 | 1.02 | 0.45 to 2.24 | 0.95 |

(*Continued on next page*)

**TABLE 1** Univariate analysis of demographic, clinical, and treatment variables between MDR and non-MDR Gram-negative bacterial BSI in cancer patients (*Continued*)

| Variables | MDR (n = 293) | MDR (%) (IQR) | N-MDR (n = 436) | N-MDR (%) (IQR) | Odds ratio | 95% CI | P value |
|---|---|---|---|---|---|---|---|
| Ascites | 40 | 13.65 | 55 | 12.61 | 1.1 | 0.71 to 1.70 | 0.68 |
| Leukopenia | 30 | 10.24 | 46 | 10.55 | 0.97 | 0.59 to 1.56 | 0.89 |
| Cerebral infarction | 7 | 2.39 | 17 | 3.9 | 0.6 | 0.23 to 1.39 | 0.26 |
| Electrolyte disorders | 67 | 22.87 | 75 | 17.2 | 1.43 | 0.98 to 2.07 | 0.06 |
| Neutropenia | 38 | 12.97 | 68 | 15.6 | 0.81 | 0.53 to 1.25 | 0.32 |
| Neutrophilia | 134 | 45.73 | 182 | 41.74 | 1.18 | 0.87 to 1.59 | 0.29 |
| Hospital stays (in days) | 22 | (12 to 32) | 18 | (12 to 30) | −2[b] | | 0.06[a] |
| Mortality (30 days) | 20 | 6.83 | 15 | 3.44 | 2.06 | 1.05 to 3.99 | **0.04** |

[a]Mann-Whitney U test.
[b]Difference: Hodges-Lehmann; CVC, central venous catheter; PICC, peripherally inserted central catheter.
[c]Bold values represent statistically significant results.

1,161 bacterial isolates were identified, comprising 62.79% Gram-negative and 37.21% Gram-positive species. Historically, BSIs were primarily caused by Gram-positive bacteria; however, recent trends indicate a rising prevalence of Gram-negative infections. This shift is attributed to factors such as increasing antibiotic resistance, the growing use of invasive procedures, and globalization (17). Among the Gram-negative isolates, *E. coli* and *Klebsiella* species were most prevalent, indicating their persistence in the hospital environment and their frequent involvement in hospital-acquired infections. These species pose a particular concern in oncology units because they can cause severe infections and often exhibit multidrug resistance (18). Among Gram-positive bacteria, *S. aureus* and *S. epidermidis* significantly contributed to the burden of BSI, highlighting the need for strict infection control measures, particularly regarding indwelling devices, which are often associated with Gram-positive infections (19).

Regarding the cancer types, both Gram-positive and Gram-negative bacteria were associated with hematological cancers and carcinoma. The proportion of Gram-positive bacteria was high in lymphoma, lung cancer, head and neck cancer, and sarcoma. This may be due to the opportunistic nature of skin flora, such as *Staphylococcus* species and *Enterococcus* species in immunocompromised patients, along with immune suppression from cancer therapies and weakened skin and mucosal barriers that allow these bacteria to cause infections (1, 20). Gram-negative bacteria were reported more frequently than Gram-positive bacteria in gastrointestinal, hepatobiliary, and breast cancers. This high proportion of Gram-negative bacteria might be due to the translocation of enteric pathogens resulting from the disruption of the gastrointestinal and biliary tracts, caused by surgeries or chemotherapy-induced mucosal damage (21, 22). Previous basic science research has suggested that breast tumor tissue may harbor a distinct microbiota enriched with Gram-negative bacteria, which may contribute to the local tumor environment and immune modulation (23). However, further studies are needed to understand the translocation mechanisms linking local Gram-negative microbiota with bloodstream infection.

Regarding gender, both Gram-negative and Gram-positive BSIs were more prevalent in males than in females. The result aligns with previously published studies, and the gender differences in bacterial BSIs may be due to hormonal differences and variations in immune response between males and females (24, 25). A higher prevalence of Gram-positive BSIs compared to Gram-negative infections was observed in patients with ventilators, intra-abdominal catheters, nasogastric feeding, and those who had undergone surgical interventions. This could be attributed to the ability of Gram-positive bacteria to colonize and form biofilms on medical devices, disrupt normal flora, and exploit compromised immune defenses (26, 27). The significantly higher rate of thrombosis in the Gram-positive BSI group in this study might be due to a stronger immune response triggered by their cell wall components, virulence factors like coagulases and protein A, increased biofilm formation, and platelet activation and aggregation (28). In contrast, hypoproteinemia and neutrophilia in Gram-negative

**TABLE 2** Univariate analysis of demographic, clinical, and treatment variables between MDR and non-MDR Gram-positive bacterial BSI in cancer patients

| Variables | MDR (n = 220) | MDR (%) (IQR) | N-MDR (n = 212) | N-MDR (%) (IQR) | Odds ratio | 95% CI | P value |
|---|---|---|---|---|---|---|---|
| Gender | | | | | | | |
| Male | 137 | 62.27 | 129 | 60.85 | 1.06 | 0.72 to 1.57 | 0.76 |
| Female | 83 | 37.73 | 83 | 39.15 | 0.94 | 0.64 to 1.39 | 0.76 |
| Age (median [IQR]) | 57 | (48 to 66) | 59 | (50 to 68) | 1[b] | | 0.42[a] |
| Occupation | | | | | | | |
| Jobless/unemployed/retired | 70 | 31.82 | 67 | 31.6 | 1.01 | 0.67 to 1.52 | 0.96 |
| Laborer/manual worker | 116 | 52.73 | 114 | 53.77 | 0.96 | 0.66 to 1.39 | 0.83 |
| Freelancer/self-employed | 19 | 8.64 | 16 | 7.55 | 1.16 | 0.58 to 2.24 | 0.68 |
| Professional and technical | 15 | 6.82 | 15 | 7.08 | 0.96 | 0.45 to 2.03 | 0.92 |
| Education background | | | | | | | |
| No formal education | 58 | 26.36 | 67 | 31.6 | 0.77 | 0.51 to 1.16 | 0.23 |
| Basic education | 66 | 30 | 62 | 29.25 | 1.04 | 0.68 to 1.55 | 0.86 |
| Intermediate education | 40 | 18.18 | 28 | 13.21 | 1.46 | 0.86 to 2.47 | 0.16 |
| Secondary education | 47 | 21.36 | 37 | 17.45 | 1.29 | 0.80 to 2.11 | 0.3 |
| Higher education | 9 | 4.09 | 18 | 8.49 | 0.46 | 0.20 to 1.04 | 0.06 |
| Source of infection | | | | | | | |
| Hospital acquired | 193 | 87.73 | 184 | 86.79 | 1.09 | 0.63 to 1.89 | 0.77 |
| Community acquired | 27 | 12.27 | 28 | 13.21 | 0.92 | 0.53 to 1.59 | 0.77 |
| Cancer treatment | | | | | | | |
| Chemotherapy | 127 | 57.73 | 126 | 59.43 | 0.93 | 0.64 to 1.37 | 0.72 |
| Radiotherapy | 21 | 9.55 | 27 | 12.74 | 0.72 | 0.40 to 1.34 | 0.29 |
| Targeted therapy | 39 | 17.73 | 49 | 23.11 | 0.72 | 0.45 to 1.13 | 0.16 |
| Immunotherapy | 10 | 4.55 | 23 | 10.85 | 0.39 | 0.19 to 0.84 | **0.01**[c] |
| Procedures and Interventions | | | | | | | |
| CVC | 34 | 15.45 | 36 | 16.98 | 0.89 | 0.53 to 1.50 | 0.67 |
| Urinary catheter | 61 | 27.73 | 40 | 18.87 | 1.65 | 1.06 to 2.57 | **0.03** |
| PICC | 42 | 19.09 | 33 | 15.57 | 1.28 | 0.79 to 2.13 | 0.33 |
| Safety IV catheter | 134 | 60.91 | 125 | 58.96 | 1.08 | 0.74 to 1.60 | 0.68 |
| Ventilator | 17 | 7.73 | 10 | 4.72 | 1.69 | 0.76 to 3.65 | 0.2 |
| Blood transfusion | 56 | 25.45 | 40 | 18.87 | 1.47 | 0.93 to 2.31 | 0.1 |
| Biopsies | 18 | 8.18 | 14 | 6.6 | 1.26 | 0.62 to 2.66 | 0.53 |
| Intra-abdominal catheter | 27 | 12.27 | 18 | 8.49 | 1.51 | 0.83 to 2.77 | 0.2 |
| Parenteral nutrition | 56 | 25.45 | 63 | 29.72 | 0.81 | 0.53 to 1.23 | 0.32 |
| Nasogastric feeding | 26 | 11.82 | 13 | 6.13 | 2.05 | 1.02 to 4.25 | **0.04** |
| Surgery | 46 | 20.91 | 34 | 16.04 | 1.38 | 0.85 to 2.25 | 0.19 |
| Infection and infectious diseases | | | | | | | |
| Sepsis | 105 | 47.73 | 104 | 49.06 | 0.95 | 0.65 to 1.37 | 0.78 |
| Septic shock | 9 | 4.09 | 7 | 3.3 | 1.25 | 0.49 to 3.26 | 0.66 |
| Hepatitis B | 28 | 12.73 | 28 | 13.21 | 0.96 | 0.56 to 1.65 | 0.88 |
| Pneumonia | 39 | 17.73 | 37 | 17.45 | 1.02 | 0.63 to 1.66 | 0.94 |
| Urinary tract infection | 13 | 5.91 | 3 | 1.42 | 4.38 | 1.26 to 14.57 | **0.01** |
| Biliary tract infection | 12 | 5.45 | 7 | 3.3 | 1.69 | 0.69 to 4.36 | 0.28 |
| Conditions and complications | | | | | | | |
| Anemia | 55 | 25 | 60 | 28.3 | 0.84 | 0.55 to 1.30 | 0.44 |
| Hypertension | 34 | 15.45 | 33 | 15.57 | 0.99 | 0.58 to 1.64 | 0.97 |
| Diabetes | 15 | 6.82 | 26 | 12.26 | 0.52 | 0.28 to 1.01 | 0.05 |
| Thrombosis | 6 | 2.73 | 19 | 8.96 | 0.28 | 0.12 to 0.68 | **0.01** |
| Myelosuppression | 64 | 29.09 | 57 | 26.89 | 1.12 | 0.74 to 1.70 | 0.61 |
| Hypoproteinemia | 77 | 35 | 60 | 28.3 | 1.36 | 0.91 to 2.07 | 0.13 |
| Nausea and vomiting | 9 | 4.09 | 2 | 0.94 | 4.48 | 1.15 to 20.83 | **0.04** |
| Radiation dermatitis | 2 | 0.91 | 3 | 1.42 | 0.64 | 0.11 to 3.16 | 0.62 |
| Respiratory failure | 14 | 6.36 | 7 | 3.3 | 1.99 | 0.78 to 5.18 | 0.14 |

**TABLE 2** Univariate analysis of demographic, clinical, and treatment variables between MDR and non-MDR Gram-positive bacterial BSI in cancer patients (*Continued*)

| Variables | MDR (*n* = 220) | MDR (%) (IQR) | N-MDR (*n* = 212) | N-MDR (%) (IQR) | Odds ratio | 95% CI | *P* value |
|---|---|---|---|---|---|---|---|
| Ascites | 26 | 11.82 | 16 | 7.55 | 1.64 | 0.86 to 3.24 | 0.13 |
| Leukopenia | 24 | 10.91 | 25 | 11.79 | 0.92 | 0.51 to 1.69 | 0.77 |
| Cerebral infarction | 6 | 2.73 | 8 | 3.77 | 0.72 | 0.24 to 2.12 | 0.54 |
| Electrolyte disorders | 40 | 18.18 | 40 | 18.87 | 0.96 | 0.58 to 1.57 | 0.85 |
| Neutropenia | 24 | 10.91 | 27 | 12.74 | 0.84 | 0.46 to 1.51 | 0.56 |
| Neutrophilia | 72 | 32.73 | 87 | 41.04 | 0.7 | 0.48 to 1.04 | 0.07 |
| Hospital stays (in days) | 21 | (13 to 31) | 20 | (14 to 32) | −1[b] | | 0.96[a] |
| Mortality (30 days) | 11 | 5 | 12 | 5.66 | 0.88 | 0.40 to 2.09 | 0.76 |

[a]Mann-Whitney U test.
[b]Difference: Hodges-Lehmann; CVC, central venous catheter; PICC, peripherally inserted central catheter.
[c]Bold values represent statistically significant results.

BSIs may be attributed to the release of endotoxin lipopolysaccharide (LPS) and other virulence factors. Endotoxins activate secondary inflammatory responses, which can increase vascular permeability, alter protein metabolism, and lower serum albumin levels. Similarly, the strong immune response triggered by exotoxins and LPS enhances neutrophil recruitment to the site of infection (29).

In the current study, high resistance rates were observed among Gram-negative bacteria against commonly used antibiotics. Specifically in *E. coli*, along with the first-line drugs, like ampicillin and cefazoline, a concerning resistance has been reported against ceftriaxone (54.39%), which is a third-generation cephalosporin. This finding aligns with a previous study from China, which reported that nearly 70% of *E. coli* strains were resistant to ceftriaxone (30). The increasing resistance to ceftriaxone is a worldwide concern. The Centers for Disease Control and Prevention reported a 53% increase in ceftriaxone resistance in clinical cultures (31). The most frequent mechanism of ceftriaxone resistance in *E. coli* is ESBL production. The proportion of ESBL-producing *E. coli* in the current study was 20.95%, consistent with a previously published study from Pakistan reporting 22.2% (32). However, other studies have reported high rates of ESBL-producing *E. coli* in clinical settings in China, with prevalence rates of 40.98% and 43.8% (33, 34). The contradictions in the results might be due to differences in geographical location and clinical practices. The carbapenemase producers are another major concern; they exhibit resistance to carbapenem drugs like meropenem and imipenem, which are considered the last resort options in clinical settings (35). In the current study, 2.69% of *E. coli* and 7.87% of *Klebsiella* species were carbapenem-resistant. The proportion of carbapenemase-producing Enterobacterales in the current study is lower than that reported in a recent study from Changsha, China, which found 15.6% carbapenem-resistant *K. pneumoniae* in clinical samples (36). However, close attention to the clinical use of carbapenem drugs is required, as plasmid-mediated carbapenem-resistant genes, such as $bla_{NDM}$ and $bla_{KPC}$, have horizontal transferability and can spread rapidly in clinical settings (15). Among all antibiotics tested against Gram-negative isolates, tigecycline was the most effective antibiotic, with only one *Klebsiella* isolate showing resistance, and all other Gram-negative bacteria remained susceptible. Although tigecycline susceptibility is recognized worldwide, some exceptional cases with

**TABLE 3** Multivariate regression analysis of risk factors for MDR Gram-negative BSIs in cancer patients

| Variables | OR (95% CI) | *P* value |
|---|---|---|
| Age | 0.98 (0.96–0.99) | 0.05 |
| Urinary catheters | 1.86 (0.95–3.70) | 0.07 |
| Urinary tract infection | 2.37 (0.91–6.67) | 0.08 |
| Parenteral nutrition | 0.72 (0.34–1.5) | 0.38 |
| Hypoproteinemia | 16.23 (6.66–46) | **<0.001**[a] |
| Mortality (30 days) | 0.57 (0.17–1.74) | 0.34 |

[a]Bold values represent statistically significant results.

higher resistance have been reported, mainly due to plasmid-mediated resistance genes like *tet* (*X*). Therefore, careful stewardship of this last-resort drug is required (37). The current study found a high level of ampicillin non-susceptibility among *Klebsiella* species, primarily attributed to their intrinsic resistance to this antibiotic. Previous studies have shown that *Klebsiella* species possess the chromosomally encoded *bla*$_{SHV-1}$ β-lactamase gene, which confers resistance to ampicillin (38). However, this resistance is not uniform across all *Klebsiella* species. Variations in the regulation and promoter mutations of the *bla*$_{SHV-1}$ gene have been reported to result in differing MIC values and phenotypic resistance profiles (38). Among the *Klebsiella* isolates tested, four strains (3.03%) were found to be susceptible to ampicillin. This also suggests that, although *Klebsiella* species are intrinsically resistant to ampicillin, clinical decisions should still be guided by susceptibility testing (39). Furthermore, the presence of intrinsic resistance underscores the need to consider alternative β-lactam combinations for the effective treatment of *Klebsiella* infections (38).

For both *S. aureus* and CoNS, more than 87% of isolates were found resistant to ampicillin and penicillin G. The resistance to these β-lactam antibiotics in our study is higher than a previously reported result from China, which found 81.6% resistance to penicillin G in *S. aureus* (40). However, another study involving isolates from hematological malignancies showed 100% resistance to penicillin G for both *S. aureus* and CoNS. This high resistance rate might be due to the development of acquired resistance genes, driven by the widespread use of β-lactam antibiotics in hospital settings, coupled with the ability of these genes to spread through horizontal gene transfer (41). Furthermore, 36.04% of *S. aureus* and 66.83% of CoNS were methicillin-resistant. The prevalence of MRSA in the current study aligns with previous studies from China, which reported MRSA rates of 29% and 34% in clinical settings. However, the prevalence of MRCoNS in the current study is substantially higher than that reported in earlier studies from China (39% to 46%) (42). The high prevalence of MRCoNS in our study highlights the importance for healthcare workers to improve infection control and hygiene practices, as CoNS are commonly transmitted within healthcare settings (43). The high resistance rate of *Enterococcus* species to ciprofloxacin and levofloxacin may be attributed to prolonged exposure and selective pressure. Frequent administration of these antimicrobials in clinical settings facilitates the adaptation of resistant strains (44).

Furthermore, risk factor analysis for MDR was conducted. Hypoproteinemia was identified as an independent risk factor for MDR Gram-negative BSIs. This finding suggests that compromised nutritional status and underlying systemic illness, which often result in low protein levels, may predispose patients to more severe infections. These findings underscore the importance of monitoring and managing hypoproteinemia to mitigate the risk of MDR development (45).

The limitations of the current study include its single-center, retrospective design and differences in patient management protocols, which may limit the generalizability of the findings to other clinical settings. However, the comprehensive analysis of bacterial BSIs in cancer patients, focused on MDR and robust microbiological methods, makes this study highly relevant to clinical practices. The findings of this study highlight the need for strengthened antimicrobial stewardship, patient-specific personalized treatment strategies, and potential revisions to empirical therapies and infection control policies. Additionally, future prospective research in more diverse clinical settings is required to confirm and build upon these findings.

## Conclusion

The current study comprehensively analyzed bacterial BSIs in cancer patients and identified *E. coli*, *Klebsiella* species, and *S. aureus* as the primary contributors to these infections. Overall, Gram-negative BSIs were more common than Gram-positive, with a higher prevalence in males, while Gram-positive BSIs were frequently associated with patients undergoing invasive procedures. High resistance rates to commonly used antibiotics, third-generation cephalosporins, and carbapenem drugs were observed.

Hypoproteinemia was identified as an independent risk factor for the emergence of MDR bacteria, suggesting the need for more precise treatment and patient-specific risk management. This study provides guidance for healthcare workers in similar settings regarding infection control. However, large-scale antimicrobial stewardship programs, multicenter studies, and prospective molecular analyses are required to develop more robust guidelines for managing BSIs in cancer patients.

## ACKNOWLEDGMENTS

We are thankful to the Jiangxi Cancer Hospital for supporting this study.

This work was supported by the Science and Technology Research Project of Jiangxi Provincial Department of Education (GJJ2403604) and (GJJ2203508); the Research start-up fund of Jiangxi Cancer Hospital (BSQDJ2024001); the Distinguished Young Scholars program of the Natural Science Foundation of Jiangxi Province (20224ACB216015), the Distinguished Young Scholars Fund of Jiangxi Cancer Hospital (2021DYS01), and by the 2023 Key Project for Science and Technology Innovation of Jiangxi Provincial Health Commission (2023ZD005).

Conceptualization: H.B. and B.X.; methodology: B.X., H.B., and X.W.; software: H.B. and X.L.; validation: R.U.K. and M.N.K.; formal analysis: M.S. and S.K.; investigation: H.B. and T.Y.; resources: X.W. and Y.S.; data curation: X.L. and H.Q.; writing—original draft preparation: H.B.; writing—review and editing: T.Y., R.U.K., B.X., X.W., X.L., M.S., S.K., Y.S., H.Q., Q.-L.L., and M.N.K.; visualization: H.B. and R.U.K.; supervision: Q.-L.L.; project administration: B.X. All authors read and agreed to the published version of the manuscript.

Patient consent was waived as we obtained data from the hospital surveillance system as a secondary source, and no patient image or figure is involved.

## AUTHOR AFFILIATIONS

[1]Jiangxi Key Laboratory of Oncology (2024SSY06041), JXHC Key Laboratory of Tumour Metastasis, NHC Key Laboratory of Personalized Diagnosis and Treatment of Nasopharyngeal Carcinoma, Jiangxi Cancer Hospital & Institute, The Second Affiliated Hospital of Nanchang Medical College, Nanchang, Jiangxi, China
[2]Department of Medical Laboratory, Jiangxi Cancer Hospital, The Second Affiliated Hospital of Nanchang Medical College, Jiangxi Cancer Institute, Nanchang, Jiangxi, China
[3]Department of Cell Biology and Genetics, Shantou University Medical College, Shantou, China
[4]Research Institute of Clinical Pharmacy, Department of Pharmacology, Shantou University Medical College, Shantou, China
[5]Department of Dermatology, The Second Affiliated Hospital of Shantou University Medical College, Shantou, China
[6]College of Life Sciences, University of Chinese Academy of Sciences, Beijing, China
[7]CAS Key Laboratory of Pathogen Microbiology and Immunology, Institute of Microbiology, Center for Influenza Research and Early-warning (CASCIRE), CAS-TWAS Center of Excellence for Emerging Infectious Diseases (CEEID), Chinese Academy of Sciences (CAS), Beijing, China

## AUTHOR ORCIDs

Muhammad Shafiq  https://orcid.org/0000-0002-4346-5903
Qiao-Li Lv  http://orcid.org/0000-0002-0452-7032
Hazrat Bilal  http://orcid.org/0000-0002-9679-3054

## AUTHOR CONTRIBUTIONS

Bin Xu, Conceptualization, Funding acquisition, Methodology, Project administration, Writing – review and editing | Xunsong Wang, Methodology, Resources, Writing – review and editing | Xiaohui Li, Data curation, Software, Writing – review and editing |

Muhammad Nadeem Khan, Validation, Writing – review and editing | Muhammad Shafiq, Formal analysis, Writing – review and editing | Sabir Khan, Formal analysis, Writing – review and editing | Tenghua Yu, Investigation, Writing – review and editing | Rahat Ullah Khan, Validation, Writing – review and editing | Ying Song, Resources, Writing – review and editing | Hanman Qiu, Data curation, Writing – review and editing | Qiao-Li Lv, Funding acquisition, Supervision, Writing – review and editing | Hazrat Bilal, Conceptualization, Funding acquisition, Investigation, Methodology, Software, Visualization, Writing – original draft, Writing – review and editing

## DATA AVAILABILITY

The raw data supporting the conclusions of this article will be made available by the authors without undue reservation.

## ETHICS APPROVAL

Ethical approval was provided by the Human Research Ethics Committee of the Jiangxi Cancer Hospital (Ref: 2024ky080) following the Declaration of Helsinki criteria. Consent forms from the patients were waived by the ethical committee, as all the data were retrospectively collected from the hospital laboratory as routine work and not for this study.

## ADDITIONAL FILES

The following material is available online.

### Supplemental Material

**Supplemental material (Spectrum01468-25-s0001.pdf).** Fig. S1; Tables S1 to S8.

### Open Peer Review

**PEER REVIEW HISTORY (review-history.pdf).** An accounting of the reviewer comments and feedback.

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
