## [Reviewer comments · Microbiology Spectrum]

Microbiology Spectrum

Six-Year Retrospective Analysis of the Epidemiology and Risk Factors of Multidrug-Resistant Bloodstream Infections in Oncology Patients in Jiangxi, China

Bin Xu, Xunsong Wang, Xiaohui Li, Muhammad Khan, Muhammad Shafiq, sabir khan, Tenghua Yu, Rahat Ullah Khan, Ying Song, Hanman Qiu, Qiao-Li Lv, and Hazrat Bilal

Corresponding Author(s): Hazrat Bilal, Jiangxi Key Laboratory of oncology, JXHC Key Laboratory of Tumor Metastasis, Jiangxi Cancer Hospital, The Second Affiliated Hospital of Nanchang Medical College, Jiangxi Cancer Institute, Nanchang, Jiangxi

Review Timeline:

Submission Date:	May 11, 2025
Editorial Decision:	June 13, 2025
Revision Received:	July 7, 2025
Editorial Decision:	July 20, 2025
Revision Received:	July 22, 2025
Editorial Decision:	July 24, 2025
Revision Received:	July 24, 2025
Accepted:	July 26, 2025

Editor: Paschalis Vergidis

Reviewer(s): Disclosure of reviewer identity is with reference to reviewer comments included in decision letter(s). The following individuals involved in review of your submission have agreed to reveal their identity: Gabriele Giuliano (Reviewer #1)

Transaction Report:

DOI: <https://doi.org/10.1128/spectrum.01468-25>

Re: Spectrum01468-25 (**Seven-Year Retrospective Analysis of the Epidemiology and Risk Factors of Multidrug-Resistant Bloodstream Infections in Oncology Patients in Jiangxi, China**)

Dear Dr. Hazrat Bilal:

Thank you for the privilege of reviewing your work. Below you will find my comments, instructions from the Spectrum editorial office, and the reviewer comments.

Editor's comments: Empiric antibiotic therapy was defined as treatment given at least 48 h before the BSI report. Treatment was deemed inappropriate if the recovered organisms were in vitro resistant to the antibiotics used. The authors concluded that inappropriate antibiotic therapy was an independent risk factor for infection.- It is well known that inappropriate antibiotic use can contribute to the emergence of resistance. However, in this study, patients were found to be bacteremic because the empirically prescribed antibiotics were ineffective against the active infection being treated. Therefore, I recommend that the authors remove "inappropriate antibiotic therapy" from their analysis.

Revision Guidelines

Sincerely,
Paschalis Vergidis
Editor
Microbiology Spectrum

Reviewer #1 (Comments for the Author):

The study aims to describe various aspects of bloodstream infections (BSIs) in oncology patients. The sample size is very large, which suggests that the findings could represent a significant contribution to the scientific literature on this topic. However, substantial revisions are needed to enhance the quality and clarity of the manuscript.

The English language throughout the manuscript requires extensive revision to improve readability and clarity.

The results section should be presented in a much clearer and more straightforward manner, as the current structure appears disorganized and difficult to follow.

Tables should be redesigned for easier comprehension, and the same applies to the figures. Moreover, both tables and figures should maintain a consistent presentation style throughout the manuscript.

In several instances, the data related to Gram-positive and Gram-negative pathogens are reported in a confusing manner. Dividing the content into separate sections for Gram-negative and Gram-positive bacteria-both in the text and in the tables/figures-could significantly enhance clarity.

Once again, overall clarity should be greatly improved, as some key information is not easily accessible to the reader. In particular, the Methods section should clearly specify the primary and secondary endpoints. There are frequent references to "risk factors" without indicating precisely what they are associated with (e.g., BSI? MDR-BSI?). Complications are mentioned but not defined.

Reviewer #2 (Comments for the Author):

1. Revise the title from "Seven-Year" to "Six-Year (2019-2024)" to accurately reflect the study period.
2. Hyphenate "Multi-drug resistance" in the keywords to "Multidrug-resistance" for consistency with standard terminology.
3. Remove the repeated definition of "BSI" in the abstract; the abbreviation alone is sufficient after its initial introduction.
4. Replace "underscoring" with "highlighting" in the abstract for smoother phrasing.
5. Revise "such chemotherapy" to "such as chemotherapy" in the Background section for grammatical accuracy.
6. Replace "abuse" with "overuse" in the phrase "widespread abuse of antibiotics" to maintain a formal and neutral tone.
7. Correct "smart MS 5020" to "MALDI-TOF MS Smart 5020" for clarity and consistency with the instrument's full name.
8. Rephrase "deemed inappropriate if no single antibiotics show in vitro sensitivity" to "deemed inappropriate if none of the administered antibiotics showed in vitro sensitivity."
9. Clarify how variables were selected for inclusion in the multivariate logistic regression analysis. Was a p-value threshold from the univariate analysis used?
10. Contextualize the high resistance of *Klebsiella* species to ampicillin as intrinsic resistance. Mention this in the Results section and elaborate on its implications in the Discussion.

Editor's comments: Empiric antibiotic therapy was defined as treatment given at least 48 h before the BSI report. Treatment was deemed inappropriate if the recovered organisms were in vitro resistant to the antibiotics used. The authors concluded that inappropriate antibiotic therapy was an independent risk factor for infection.- It is well known that inappropriate antibiotic use can contribute to the emergence of resistance. However, in this study, patients were found to be bacteremic because the empirically prescribed antibiotics were ineffective against the active infection being treated. Therefore, I recommend that the authors remove "inappropriate antibiotic therapy" from their analysis.

Response: We sincerely thank the Editor for this important observation. Following the recommendation, we have removed the analysis of "inappropriate antibiotic therapy" from the manuscript. We appreciate your guidance in strengthening the focus of our study.

Thank you!

Reviewer #1

Comment # 1: The study aims to describe various aspects of bloodstream infections (BSIs) in oncology patients. The sample size is very large, which suggests that the findings could represent a significant contribution to the scientific literature on this topic. However, substantial revisions are needed to enhance the quality and clarity of the manuscript. The English language throughout the manuscript requires extensive revision to improve readability and clarity.

Response: We sincerely thank the Editor for the valuable feedback and for recognizing the potential contribution of our study. We fully appreciate the importance of clarity and readability. In response to the suggestion, we have thoroughly revised

the manuscript to improve the quality of the English language. The revised version has been carefully edited by a native English speaker with expertise in scientific writing to ensure clarity, fluency, and precision. We hope that these revisions meet the journal's expectations.

Comment # 2: The results section should be presented in a much clearer and more straightforward manner, as the current structure appears disorganized and difficult to follow.

Response: We appreciate the reviewer's valuable suggestion regarding the need to present the Results section more clearly and straightforwardly. In response, we have made substantial revisions to improve the structure and clarity of this section. Specifically, we added a new subtitle to the first paragraph (*3.1 Overview of BSI Episodes and Bacterial Isolates*). In this section, we first present the results of BSI episodes in oncological patients, followed by the distribution of bacterial types, and then detail the prevalence of gram-positive and gram-negative bacteria across various cancer types. The data on MDR versus non-MDR in various cancer types have been relocated from this section to a newly designated section (*3.4: MDR versus non-MDR*).

Furthermore, we modified the subtitle of the demographic and clinical characteristics section to (*3.2 Comparative Analysis of Demographic and Clinical Characteristics Between Gram-Negative and Gram-Positive Bacterial Cases*), where we comparatively analyzed all study characteristics between gram-positive and gram-negative cases. Section *3.3 Antibiotic Susceptibility Profiles* is now clearly divided into subsections for gram-negative and gram-positive bacteria, each with its subtitle for easier understanding.

Finally, (3.4 *MDR versus non-MDR*) has been reorganized into three subsections: (3.4.1 *Risk Factors for MDR in Gram-Negative Bacterial BSIs*), (3.4.2 *Risk Factors for MDR in Gram-Positive Bacterial BSIs*), (each presenting specific risk factors), and 3.4.3 *Multivariate Regression Analysis*, where we report multivariate regression analyses for both gram-negative and gram-positive cases. We trust these changes have substantially improved the clarity and organization of the "Results" section.

Comment # 3: Tables should be redesigned for easier comprehension, and the same applies to the figures. Moreover, both tables and figures should maintain a consistent presentation style throughout the manuscript.

Response: We appreciate the reviewer's valuable suggestion regarding the redesign and consistency of tables and figures. In response, we have substantially revised these elements to enhance clarity and readability. Specifically, the original Figure 1 has been divided into two separate figures: the new **Figure 1**, which presents only the comparison of gram-positive and gram-negative cases across various cancer types and is now placed in **Section 3.2: Comparative analysis of demographic and clinical characteristics between gram-negative and gram-positive bacterial cases**; and **Figure 4**, which illustrates the comparative distribution of MDR versus non-MDR cases in both gram-negative and gram-positive infections across cancer types, now placed in **Section 3.4: MDR versus non-MDR**.

Regarding the tables, we have included **three main tables** in the manuscript. **Table 1** presents the univariate analysis of demographic, clinical, and treatment variables between MDR and non-MDR gram-negative bacterial BSI in cancer patients. **Table 2** presents the corresponding univariate analysis for gram-positive cases. To ensure readability, we organized each table into clearly defined subsections for different

demographic and clinical variables rather than splitting them into multiple smaller tables, which would have resulted in an excessive number of tables and reduced comprehension. Finally, **Table 3** presents the multivariate regression analysis of MDR in gram-positive and gram-negative BSI, with distinct sections for each group. We have ensured that the style and format of all tables and figures are consistent throughout the manuscript.

Comment # 4: In several instances, the data related to Gram-positive and Gram-negative pathogens are reported in a confusing manner. Dividing the content into separate sections for Gram-negative and Gram-positive bacteria-both in the text and in the tables/figures-could significantly enhance clarity.

Response: We appreciate the reviewer's thoughtful feedback. To improve clarity, we have structured the results as follows: **Section 3.3 (Antibiotic Susceptibility Profiles) is now divided into 3.3.1 AST of Gram-negative Bacteria and 3.3.2 AST of Gram-positive Bacteria.** Similarly, **Section 3.4 (MDR versus non-MDR)** is divided into **two subsections: 3.4.1, "Risk factors for MDR in Gram-negative bacterial BSIs," and 3.4.2, "Risk factors for MDR in Gram-positive bacterial BSIs."**

Regarding **Section 3.2 (Comparative analysis of demographic and clinical characteristics between Gram-negative and Gram-positive bacterial cases)**, as this section provides a direct comparison, it could not be split into separate parts. However, to avoid confusion, we have added clarifying statements such as: *"Both hospital-acquired and community-acquired cases were significantly more common in gram-negative BSIs than in gram-positive BSIs ($p < 0.001$)."* and *"The difference between gram-positive and gram-negative bacterial groups across various cancer*

treatments was not statistically significant."

For the tables, Table 1 focuses solely on gram-negative cases, Table 2 on gram-positive cases, and Table 3 presents both, with clearly divided sections: gram-negative data on the left and gram-positive data on the right, each section having explicit titles.

Regarding figures: **Figure 1** provides a comparative analysis of gram-positive and gram-negative cases, which could not be separated; however, distinct colors are used to differentiate the two groups. **Figure 2** shows only gram-negative isolates, **Figure 3** only gram-positive isolates, and **Figure 4** is divided into two panels (A for gram-negative, B for gram-positive).

We hope these revisions address the reviewer's concerns.

Comment # 4: Once again, overall clarity should be greatly improved, as some key information is not easily accessible to the reader. In particular, the Methods section should clearly specify the primary and secondary endpoints. There are frequent references to "risk factors" without indicating precisely what they are associated with (e.g., BSI? MDR-BSI?). Complications are mentioned but not defined.

Response: We appreciate the reviewer's valuable feedback. To improve clarity and address the concerns raised, we have made the following revisions to the Methods section:

We added a new subsection titled "*2.5 Study endpoints*" to clearly define the study objectives:

"The primary endpoint of this study was to identify risk factors associated with MDR-BSIs among cancer patients. The secondary endpoints included (1) comparison of demographic, clinical, and treatment characteristics between gram-negative and gram-

positive bacteria, (2) analysis of antibiotic susceptibility profiles, and (3) 30-day mortality following BSI diagnosis."

To specify what our risk factor analysis aimed to identify, we added the sentence:

"Risk factor analysis was conducted using univariate and multivariate logistic regression to identify variables associated with MDR-BSIs in cancer patients. Variables with a p-value < 0.10 in the univariate analysis were included in the multivariate logistic regression model.." in Section 2.4 "Data analysis".

We also addressed the comment regarding complications by adding a clear definition in Section 2.2 Definitions:

"Complications of BSI included sepsis, septic shock, secondary infections (e.g., pneumonia and urinary tract infections), organ dysfunctions such as respiratory failure and electrolyte imbalances, vascular complications like thrombosis, and treatment-related conditions including myelosuppression and neutropenia."

We hope these revisions enhance the overall clarity and accessibility of the Methods section.

Thank you!

Reviewer #2:

Comment # 1: Revise the title from "Seven-Year" to "Six-Year (2019-2024)" to accurately reflect the study period.

Response: We appreciate the reviewer's helpful suggestion. In line with this recommendation, we have revised the title to accurately reflect the study period. The

new title is:

"Six-Year Retrospective Analysis of the Epidemiology and Risk Factors of Multidrug-Resistant Bloodstream Infections in Oncology Patients in Jiangxi, China."

Comment # 2: Hyphenate "Multi-drug resistance" in the keywords to "Multidrug-resistance" for consistency with standard terminology.

Response: We appreciate the reviewer's helpful suggestion. We have revised the keywords to hyphenate the term as "Multidrug-resistance" for consistency with standard terminology.

Comment # 3. Remove the repeated definition of "BSI" in the abstract; the abbreviation alone is sufficient after its initial introduction.

Response: We thank the reviewer for bringing this to our attention. We have removed the repeated definition of "BSI" in the abstract and retained the abbreviation alone after its initial introduction.

Comment # 4. Replace "underscoring" with "highlighting" in the abstract for smoother phrasing.

Response: We appreciate the reviewer's helpful suggestion. We have replaced "underscoring" with "highlighting" and revised the sentence for grammatical accuracy. The updated sentence now reads:

"The high prevalence of MDR bacteria in BSIs among cancer patients highlights the necessity of individualized treatment and continuous monitoring in oncology settings."

Comment # 5. Revise "such chemotherapy" to "such as chemotherapy" in the Background section for grammatical accuracy.

Response: We appreciate the reviewer's careful observation. We have revised the phrase from "such chemotherapy" to "such as chemotherapy" for grammatical accuracy. The updated sentence now reads:

" The risk of BSI in cancer patients is increased by invasive treatments and procedures such as chemotherapy, central venous catheterization (CVC), and surgeries."

Comment # 6. Replace "abuse" with "overuse" in the phrase "widespread abuse of antibiotics" to maintain a formal and neutral tone.

Response: We appreciate the reviewer's helpful suggestion. We have replaced "**abuse**" with "**overuse**" to maintain a formal and neutral tone. The revised sentence now reads:

" However, the widespread overuse of antibiotics has led to the emergence of resistant bacterial strains."

Comment # 7. Correct "smart MS 5020" to "MALDI-TOF MS Smart 5020" for clarity and consistency with the instrument's full name.

Response: We appreciate the reviewer's helpful suggestion. We have corrected the instrument name for clarity and consistency. The revised sentence now reads:

" Bacterial identification was performed using MALDI-TOF MS Smart 5020."

Comment # 8. Rephrase "deemed inappropriate if no single antibiotics show in vitro sensitivity" to "deemed inappropriate if none of the administered antibiotics showed in vitro sensitivity."

Response: We appreciate the reviewer's helpful suggestion. In line with the Editor's recommendation to remove "inappropriate antibiotic therapy" from our analysis, we have omitted this definition from the manuscript.

Comment # 9. Clarify how variables were selected for inclusion in the multivariate logistic regression analysis. Was a p-value threshold from the univariate analysis used?

Response: We thank the reviewer for this important observation. To address this, we have clarified the method for variable selection in the multivariate logistic regression analysis. We added the following sentence in Section 2.4 “Data analysis”:

"Variables with a p-value < 0.10 in the univariate analysis were included in the multivariate logistic regression model."

This ensures transparency regarding how variables were chosen for inclusion in the multivariate model.

Comment # 10. Contextualize the high resistance of *Klebsiella* species to ampicillin as intrinsic resistance. Mention this in the Results section and elaborate on its implications in the Discussion.

Response: Thank you for your valuable suggestion. We have revised the manuscript accordingly. In the Results section, we updated the sentence to:

*"Among *Klebsiella* species, high resistance to ampicillin (n = 117, 88.64%) was observed, attributable to intrinsic resistance."*

Additionally, in the Discussion section, we have elaborated on the implications of this intrinsic resistance as follows:

*"The current study found a high level of ampicillin non-susceptibility among *Klebsiella* species, primarily attributed to their intrinsic resistance to this antibiotic. Previous studies have shown that *Klebsiella* species possess the chromosomally encoded *blaSHV-1* β -lactamase gene, which confers resistance to ampicillin (40).*

However, this resistance is not uniform across all Klebsiella species. Variations in the regulation and promoter mutations of the blaSHV-1 gene have been reported to result in differing MIC values and phenotypic resistance profiles (40). Among the Klebsiella isolates tested, 4 strains (3.03%) were found to be susceptible to ampicillin. This also suggests that, although Klebsiella species are intrinsically resistant to ampicillin, clinical decisions should still be guided by susceptibility testing (41). Furthermore, the presence of intrinsic resistance underscores the need to consider alternative β -lactam combinations for the effective treatment of Klebsiella infections (40)."

We hope these clarifications adequately address the comment.

Thank you!

Re: Spectrum01468-25R1 (**Six-Year Retrospective Analysis of the Epidemiology and Risk Factors of Multidrug-Resistant Bloodstream Infections in Oncology Patients in Jiangxi, China**)

Dear Dr. Hazrat Bilal:

Thank you for the privilege of reviewing your work. Below you will find my comments and instructions from the Spectrum editorial office.

I would like to thank the authors for thoroughly revising the manuscript in response to the reviewers' comments. Before the manuscript can be considered for publication, I suggest that you address the following minor points:

- By "high resistance", I assume you mean "high resistance rates". Please edit in abstract and throughout the manuscript.
- Abstract: "Tigecycline and nitrofurantoin showed good susceptibility". Please rephrase.
- Importance: Change study period from seven to six years.
- 2.2: Were pneumonia and urinary tract infection complications of BSI, or were they the source?
- 2.2: Please define neutrophilia (as you did for neutropenia).
- 2.3: Did you interpret susceptibilities according to both EUCAST and CLSI? Please clarify.
- 2.4: Primary endpoint is the main outcome that is measured to determine whether an intervention being tested is effective. As this is not an interventional study, I suggest changing to "primary aim of the study."
- 3.1: There are various types of "carcinomas" and it is unclear whether these can be classified as a single entity.
- 4 Discussion: Ref. 24 pertains to a basic science study on breast microbiota. I am not sure this explains your findings on gram-negative BSI.
- Limitations: "variability in patient management protocol, which may differ from other clinical settings". Please rephrase.
- Table 3. Correct OR (95% CI) for Gram-positive BSI.

Please return the manuscript within 30 days; if you cannot complete the modification within this time period, please contact me. If you do not wish to modify the manuscript and prefer to submit it to another journal, notify me immediately so that the manuscript may be formally withdrawn from consideration by Spectrum.

Revision Guidelines

Data availability: ASM policy requires that data be available to the public upon online posting of the article, so please verify all links to sequence records, if present, and make sure that each number retrieves the full record of the data. If a new accession number is not linked or a link is broken, provide Spectrum production staff with the correct URL for the record. If the accession numbers for new data are not publicly accessible before the expected online posting of the article, publication may be delayed;

please contact production staff (Spectrum@asmusa.org) immediately with the expected release date.

Sincerely,
Paschalis Vergidis
Editor
Microbiology Spectrum

Response to comments

Spectrum01468-25R1 (Six-Year Retrospective Analysis of the Epidemiology and Risk Factors of Multidrug-Resistant Bloodstream Infections in Oncology Patients in Jiangxi, China)

Comment # 1: By "high resistance", I assume you mean "high resistance rates". Please edit in abstract and throughout the manuscript.

Response: Thank you for pointing this out. We have carefully revised the manuscript and replaced all instances of “high resistance” with “high resistance rates” in the Abstract and throughout the text, to ensure consistency and clarity in terminology.

Comment #2: Abstract: "Tigecycline and nitrofurantoin showed good susceptibility". Please rephrase.

Response: Thank you for your valuable suggestion. We have revised the sentence in the Abstract to enhance clarity and scientific accuracy. The updated sentence now reads:

“High resistance rates were observed against ampicillin, piperacillin, cefazolin, and erythromycin, whereas tigecycline and nitrofurantoin exhibited low resistance rates among the tested bacterial isolates.”

We believe this revised phrasing more precisely reflects the data and maintains a formal scientific tone.

Comment # 3: Importance: Change study period from seven to six years.

Response: Thank you for your observation. We have corrected the study period from seven to six years as suggested.

Comment # 4: 2.2: Were pneumonia and urinary tract infection complications of BSI, or were they the source?

Response: Thank you for your insightful comment. Upon review, we acknowledge that pneumonia and urinary tract infections were more likely co-infections or potential sources of BSI, rather than direct complications. To address this, we have removed them from the list of complications and included a separate sentence to clarify their role as co-infections or possible sources. The revised text reads:

"Complications of BSI included sepsis, septic shock, organ dysfunctions such as respiratory failure and electrolyte imbalances, vascular complications like thrombosis, and treatment-related conditions including myelosuppression and neutropenia. Co-infections or potential sources of BSI included pneumonia, urinary tract infections (UTI), and biliary tract infections."

Comment # 5: 2.2: Please define neutrophilia (as you did for neutropenia).

Response: Thank you for your comment. We have added a definition for neutrophilia to ensure consistency and clarity. The revised sentence now reads:

"An absolute neutrophil count below $0.5 \times 10^9/L$ is defined as neutropenia, while a count exceeding $7.5 \times 10^9/L$ is considered neutrophilia."

Comment # 6: 2.3: Did you interpret susceptibilities according to both EUCAST and CLSI? Please clarify.

Response: Thank you for your valuable comment. We have clarified this point in the revised manuscript. Susceptibility testing was primarily interpreted according to CLSI guidelines. However, for tigecycline, the CLSI breakpoints were unavailable, so we used EUCAST breakpoints (2024) for interpretation. This has now been clearly stated

in the revised text. The revised text reads:

“Sensitivity results were primarily interpreted according to CLSI guidelines for all tested antibiotics (14), except for tigecycline, which was interpreted according to EUCAST (2024) breakpoints (http://www.eucast.org/clinical_breakpoints/).”

Comment # 7: 2.4: Primary endpoint is the main outcome that is measured to determine whether an intervention being tested is effective. As this is not an interventional study, I suggest changing to "primary aim of the study."

Response: Thank you for your valuable suggestion. We agree with your observation and have revised the section title and wording accordingly. The term “primary endpoint” has been replaced with “primary aim,” and “secondary endpoints” with “secondary aims” to reflect the observational nature of the study. The revised section now reads:

“The primary aim of this study was to identify risk factors associated with MDR-BSIs among cancer patients. The secondary aims included: (1) comparison of demographic, clinical, and treatment characteristics between gram-negative and gram-positive bacteria, (2) analysis of antibiotic susceptibility profiles, and (3) assessment of 30-day mortality following BSI diagnosis.”

Comment # 8: 3.1: There are various types of "carcinomas" and it is unclear whether these can be classified as a single entity.

Response: We appreciate the reviewer’s observation. We agree that “carcinoma” encompasses a broad group of cancers, and it is important to specify the subtypes rather than treat them as a single entity. In response, we have revised the text to clarify the cancer types analyzed and have presented the data accordingly for individual cancer categories. The revised sentence now reads:

“Regarding cancer types, gram-positive bacteria were significantly more common than gram-negative bacteria in lymphoma (14.35% vs. 8.50%, $p = 0.001$), lung (13.65% vs. 9.87%, $p = 0.04$), and head and neck cancers (11.80% vs. 5.76%, $p < 0.001$). In contrast, gram-negative bacteria were significantly more prevalent than gram-positive bacteria in breast (10.56% vs. 6.71%, $p = 0.02$), hepatobiliary (20.16% vs. 13.19%, $p = 0.002$), pancreatic (5.07% vs. 2.55%, $p = 0.03$), and gastrointestinal cancers (22.91% vs. 15.97%, $p = 0.004$) (Figure 1).”

Comment # 9: 4 Discussion: Ref. 24 pertains to a basic science study on breast microbiota. I am not sure this explains your findings on gram-negative BSI.

Response: Thank you for your valuable comment. We agree that Ref. 24 is a basic science study focused on the microbiota within breast tumor tissue and does not directly explain our clinical findings on gram-negative bloodstream infections. To address this, we have revised the relevant text in the discussion section to clarify its context. The modified sentence now reads:

“Previous basic science research has suggested that breast tumor tissue may harbor a distinct microbiota enriched with gram-negative bacteria, which may contribute to the local tumor environment and immune modulation (24). However, further studies are needed to understand the translocation mechanisms linking local gram-negative microbiota with bloodstream infection.”

This revision aims to acknowledge the limitations of the reference while still providing context for potential microbiota-related mechanisms that warrant further investigation.

Comment # 10: -Limitations: "variability in patient management protocol, which may differ from other clinical settings". Please rephrase.

Response: We thank the reviewer for pointing this out. The sentence has been rephrased for improved clarity. The revised version now reads:

“The limitations of the current study include its single-center, retrospective design and differences in patient management protocols, which may limit the generalizability of the findings to other clinical settings.”

Comment # 11: Table 3. Correct OR (95% CI) for Gram-positive BSI.

Response: Thank you for pointing this out. We carefully reviewed and corrected the odds ratios (OR) and 95% confidence intervals (CI) for the Gram-positive bloodstream infections (BSI). The multivariate logistic regression analysis was repeated three times to ensure consistency and accuracy. We have updated Table 3 accordingly, which now presents the final and corrected results for both Gram-positive and Gram-negative BSI in cancer patients. The corrected Table 3 has been included in the revised manuscript as shown below:

Table 3: Multivariate regression analysis of MDR in gram positive and gram-negative BSI in cancer patients.

Gram positive			Gram negative		
Variables	OR (95%CI)	P-value	Variables	OR (95%CI)	P-value
Immunotherapy	0.00 (0.00 – 1.83×10 ¹²⁶)	0.98	Age	0.98 (0.96 – 0.99)	0.05
Urinary catheters	2.34 (1.26- 4.49)	0.008	Urinary catheters	1.86 (0.95 – 3.70)	0.07
Urinary tract infection	~7.67×10 ²⁷ (0.00 – ∞)	0.98	Urinary tract infection	2.37 (0.91 – 6.67)	0.08
Nausea and vomiting	0.00 (NA – 1.52×10 ²⁷)	0.98	Parenteral nutrition	0.72 (0.34 – 1.5)	0.38

Nasogastric feeding	$\sim 1.22 \times 10^{28}$ ($1.64 \times 10^{38} - \infty$)	0.98	Hypoproteinemia	16.23 (6.66 - 46)	<0.001
Thrombosis	0.00 (NA - 1.01×10^{81})	0.99	Mortality (30 days)	0.57 (0.17 - 1.74)	0.34

Note: Some variables in the gram-positive regression model exhibit extreme odds ratios and unusually wide confidence intervals. This is due to the small number of events and data sparsity in certain categories.

Re: Spectrum01468-25R2 (**Six-Year Retrospective Analysis of the Epidemiology and Risk Factors of Multidrug-Resistant Bloodstream Infections in Oncology Patients in Jiangxi, China**)

Dear Dr. Hazrat Bilal:

Thank you for the privilege of reviewing your work. Below you will find my comments.

Due to the limited number of events, the multivariate analysis on risk factors for multidrug resistance in Gram-positive organisms has very wide confidence intervals, rendering it uninterpretable. After reviewing with our senior editor, I recommend that the authors remove this portion of the analysis.

Please return the manuscript within 30 days; if you cannot complete the modification within this time period, please contact me. If you do not wish to modify the manuscript and prefer to submit it to another journal, notify me immediately so that the manuscript may be formally withdrawn from consideration by Spectrum.

Revision Guidelines

Sincerely,
Paschalis Vergidis
Editor
Microbiology Spectrum

Response to comments

Spectrum01468-25R2 (Six-Year Retrospective Analysis of the Epidemiology and Risk Factors of Multidrug-Resistant Bloodstream Infections in Oncology Patients in Jiangxi, China)

Comment # 1: Due to the limited number of events, the multivariate analysis on risk factors for multidrug resistance in Gram-positive organisms has very wide confidence intervals, rendering it uninterpretable. After reviewing with our senior editor, I recommend that the authors remove this portion of the analysis.

Response: We sincerely appreciate the reviewer's insightful comment. We acknowledge that the limited number of Gram-positive MDR events resulted in extremely wide confidence intervals in the multivariate analysis, making these results uninterpretable. In response, we have removed the Gram-positive multivariate analysis from the abstract, results, discussion, and conclusion sections. Only descriptive/univariate findings for Gram-positive MDR have been retained. The multivariate analysis for Gram-negative organisms remains included, as its statistical outputs were robust and clinically interpretable.

Thank you!

Re: Spectrum01468-25R3 (**Six-Year Retrospective Analysis of the Epidemiology and Risk Factors of Multidrug-Resistant Bloodstream Infections in Oncology Patients in Jiangxi, China**)

Dear Dr. Hazrat Bilal:

Your manuscript has been accepted, and I am forwarding it to the ASM production staff for publication. Your paper will first be checked to make sure all elements meet the technical requirements. ASM staff will contact you if anything needs to be revised before copyediting and production can begin. Otherwise, you will be notified when your proofs are ready to be viewed.

Sincerely,
Paschalis Vergidis
Editor
Microbiology Spectrum